# Differential impact of environmental factors on airborne live bacteria and inorganic particles in an underground walkway

**Hiroyuki Yamaguchi**[1]*, **Torahiko Okubo**[1], **Eriko Nozaki**[2], **Takako Osaki**[2]

**1** Department of Medical Laboratory Science, Faculty of Health Sciences, Hokkaido University, Sapporo, Japan, **2** Department of Infectious Diseases, Kyorin University School of Medicine, Shinkawa, Mitaka, Tokyo, Japan

* hiroyuki@med.hokudai.ac.jp

**Data Availability Statement:** All relevant data are within the manuscript and its Supporting information files.

## Abstract

We previously reported that variations in the number and type of bacteria found in public spaces are influenced by environmental factors. However, based on field survey data alone, whether the dynamics of bacteria in the air change as a result of a single environmental factor or multiple factors working together remains unclear. To address this, mathematical modeling may be applied. We therefore conducted a reanalysis of the previously acquired data using principal component analysis (PCA) in conjunction with a generalized linear model (Glm2) and a statistical analysis of variance (ANOVA) test employing the $\chi^2$ distribution. The data used for the analysis were reused from a previous public environmental survey conducted at 8:00–20:00 on May 2, June 1, and July 5, 2016 (regular sampling) and at 5:50–7:50 and 20:15–24:15 on July 17, 2017 (baseline sampling) in the Sapporo underground walking space, a 520-meter-long underground walkway. The dataset consisted of 60 samples (22 samples for "bacterial flora"), including variables such as "temperature (T)," "humidity (H)," "atmospheric pressure (A)," "traffic pedestrians (TP)," "number of inorganic particles ($\Delta5$: 1–5 μm)," "number of live airborne bacteria," and "bacterial flora." Our PCA with these environmental factors (T, H, A, and TP) revealed that the 60 samples could be categorized into four groups (G1 to G4), primarily based on variations in PC1 [Loadings: T (0.62), H(0.647), TP(0.399), A(0.196)] and PC2 [Loadings: A(0.825), TP(0.501), H(0.209), T (0.155)]. Notably, the number of inorganic particles significantly increased from G4 to G1, but the count of live bacteria was highest in G2, with no other clear pattern. Further analysis with Glm2 indicated that changes in inorganic particles could largely be explained by two variables (H/TP), while live bacteria levels were influenced by all explanatory variables (TP/A/H/T). ANOVA tests confirmed that inorganic particles and live bacteria were influenced by different factors. Moreover, there were minimal changes in bacterial flora observed among the groups (G1–G4). In conclusion, our findings suggest that the dynamics of live bacteria in the underground walkway differ from those of inorganic particles and are regulated in a complex manner by multiple environmental factors. This discovery may contribute to improving public health in urban settings.

**Funding:** This research is supported by a grant-in-aid for scientific research, KAKENHI (grant number 20K20613 to HY). The funders had no role in study design, data collection and analysis, decision to publish, or manuscript preparation.

**Competing interests:** The authors have declared that no competing interests exist.

## Introduction

Humans spend over 90% of their time indoors, in places such as homes, offices, schools, hospitals, and public areas [1–3]. Therefore, keeping these spaces clean and free from air pollution is essential, and can be confirmed through measuring the concentrations of contaminants such as heavy metals or various chemicals (benzene, toluene, ethylbenzene, or xylenes) [4–6]. Additionally, humans emit a significant number of bacteria through activities such as coughing, sneezing, talking, and breathing, contributing to indoor airborne bacteria. This can impact health by spreading or worsening infectious diseases [7–13]. Researchers have monitored microbial communities in various indoor public places to understand their dynamics [14], but our understanding of the influence of factors such as temperature, humidity, atmospheric pressure, traffic pedestrians, and inorganic particles on bacteria levels in these spaces is limited.

To address this, we previously investigated the impact of walker occupancy combined with other factors (temperature, humidity, atmospheric pressure, dust particles) on airborne bacterial features [colony forming units (CFUs) and operational taxonomic units (OTUs)] in the Sapporo Underground Pedestrian Space in Sapporo, Japan (https://www.sapporo-chikamichi.jp/) [15]. The results were interesting and revealed a positive relationship between walker occupancy and airborne bacteria that changed with increased temperature and humidity, and these findings had implications for improving public health in urban communities [15]. It has further been revealed that multiple environmental factors influence the dynamics of airborne bacteria in a complex manner, but whether this effect is based on one specific factor or multiple factors working together is unclear.

On the basis of recent research, the influence of individual environmental factors on airborne bacteria is becoming increasingly evident. For example, in a hospital room maintained at constant temperature and humidity (25˚C and 55%), similar types of fungi and bacteria were identified over at least 3 days from air samples [16]. Furthermore, the study of airborne bacteria in patient rooms showed that the diversity and composition of the indoor bacterial communities changed readily in response to variations in ventilation or temperature [17]. Other studies have shown that in the air in public spaces, the number of dust particles with attached bacteria is significantly influenced by walker occupancy [10–12]. Our previous study revealed that the dynamics of airborne bacteria (mainly derived from soil) in outdoor spaces can significantly change depending on humidity, rainfall, wind speed, and/or sunlight [18]. However, based on field survey data alone, whether the dynamics of bacteria in the air are changing due to the influence of a single environmental factor or as the result of multiple factors working together remains unclear. Mathematical modeling may be applied to address this issue.

In this study, we therefore conducted a reanalysis of the previously acquired dataset [15] using principal component analysis (PCA) in conjunction with a generalized linear model (Glm2) and performed a statistical ANOVA test employing the $\chi^2$ distribution.

## Materials and methods

### Dataset reused from our previous study and research flows

The dataset used in this study is a continuation of our previous research [15], including 60 samples (22 samples for "bacterial flora") and variables such as "temperature (T)," "humidity (H)," "atmospheric pressure (A)," "traffic pedestrians (TP)," "number of inorganic particles (Δ5: 1–5 μm)," "number of live airborne bacteria" (S1 Table), and "bacterial flora" (S2 Table); samples were collected on May 2, June 1, and July 5, 2016 (8:00 h to 20:00 h), and July 15, 2017 (5:50 h to 7:50 h / 22:15 h to 24:45 h). The analysis, as depicted in Fig 1, involved several steps. First, the dataset (T, H, A, and TP) was subjected to PCA of group variables (Fig 1, right flow).

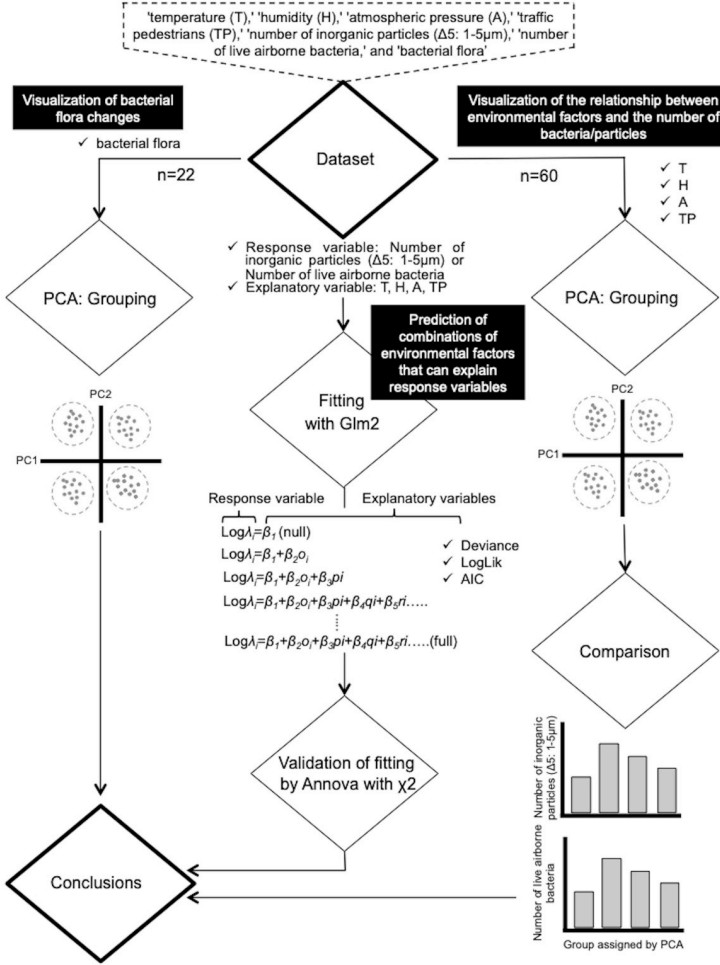

**Fig 1. Flowchart showing the research flow for the analysis.** The area surrounded by the dotted line at the top shows the contents of the dataset used, including 60 samples (22 samples for "bacterial flora") and variables with "temperature (T)," "humidity (H)," "atmospheric pressure (A)," "traffic pedestrians (TP)," "number of inorganic particles (Δ5: 1–5 μm)," "number of live airborne bacteria," and "bacterial flora" (see S1 and S2 Tables) [15]. Right flow shows the protocol for grouping the relationship between environmental factors and the number of bacteria/particles with PCA. Middle flow shows the protocol of Glm2 fitting with ANOVA using $\chi^2$. Left flow shows the protocol for grouping bacterial flora with PCA.

Subsequently, differences in viable bacterial counts and Δ5 between groups were compared. Then, Glm2 analysis, which is an R package based on "Poisson distribution," was employed to assess the degree of fit for environmental factors explaining variations in the number of bacteria and particles between groups (Fig 1, middle flow). The validity of the fitting was confirmed through a statistical ANOVA test employing the $\chi^2$ distribution. In addition, variation in the bacterial flora dataset was visualized using PCA (Fig 1, left flow). Further details for each analysis method are provided below.

## Computational performance and analysis software

A Mac computer was used [iMac (27-inch, Late 2013)] with the operating system "Mojave (version 10.14.6)." R was used for PCA, fitting with Glm2, and validation of fitting by ANOVA with $\chi^2$ (see below). R [version 4.2.3 (2023-03-15)] was run with RStudio (2021.09.1+372).

## PCA

Two analyses were performed, as shown in Fig 1 (see above). In particular, the commands below were used to determine the three values ("Standard Deviation," "Proportion of Variance," "Cumulative Proportion"). The allocation rate [Principal component (PC) 1 and 2] from the "Cumulative Proportion" to the XY axis was calculated. The degree of factors contributing to PC1 and PC2 was calculated using the command "summary." Furthermore, the contribution rate for each data point was visualized using the command "plot." In the provided R code, the temporary name "Data name.csv" was read into the variable "data." PCA was then performed using the "*princomp*" function, and a summary of the results, including the loadings, was displayed. The scores were plotted, showing the contribution rate for each data point. Details of the command line for the PCA are described in the section "Commands".

## Model fitting

As mentioned above, to assess the degree of fit for environmental factors explaining variations in the number of bacteria and particles between datasets, model fitting was run with the command "gml2." In the provided R code, "gml2" was used to model the relationship between the number of bacteria and particles and different sets of environmental factors. Three models were fitted: null control, selective combination of environmental factors, and full control. The results were displayed, and the log likelihood for each model was calculated. Finally, a comparison of the models was performed with the "fit_f" variable. From these calculations, the values of "Deviance," "logLik," and "AIC (Akaike Information Criterion)" were obtained. Details of the command line for the model fitting are described in the section "Commands".

## Validation of fitting

Validation of the degree of fitting was conducted by ANOVA with $\chi^2$. A *p*-value of 0.05 or less was considered statistically significant. Details of the command line for the validation are described in the section "Commands".

## Commands

### For PCA.

```
>data <- read.csv("Data name.csv") # "data name" is a temporary
placeholder.
>data
>result <- princomp(data, cor = TRUE)
>summary(result, loadings = TRUE)
>plot(result$scores[, 1] ~ result$scores[, 2])
>result$scores
```

### Model fitting.

```
# Null control
>fit_null <- glm2(d[, c('number of bacteria', 'particles')] ~ 1,
family = poisson)
# Selective combination of environmental factors
>fit_selective <- glm2(d[, c('number of bacteria', 'particles')] ~ d[,
'selective environmental factors'], family = poisson)
# Full control
>fit_full <- glm2(d[, c('number of bacteria', 'particles')] ~ d[, 'All
environmental factors'], family = poisson)
# Display results
>fit_null
>summary(fit_null)
```

```
# Display log likelihood for each model
>logLik(fit_null)
>logLik(fit_selective)
>logLik(fit_full)
# Comparison of models
>fit_f <- glm2(null_control, selective_combination, full_control)
>fit_f
```

**Validation of fitting.**

```
# Null control
>fit1_null<- glm2(d[, c('number of bacteria', 'particles')] ~ 1,
family = poisson)
# Selective combination of environmental factors
>fit 'selective environmental factors' <- glm2(d[, c('number of bacte-
ria', 'particles')] ~ d[, 'selective environmental factors'],
family = poisson)
# Full control
>fit_full <- glm2(d[, c('number of bacteria', 'particles')] ~ d[, 'All
environmental factors'], family = poisson)
>annova (fit_null, fit_'selective environmental factors', or fit_-
full, test = "Chisq")
```

## Other statistics

Comparisons among groups were performed by a Bonferroni–Dunn test. A $p$-value of $< 0.05$ was considered statistically significant.

## Results

### Environmental factors varied depending on the date of sample acquisition, and live bacterial counts and particulate counts showed different dynamics

First, we visualized the characteristic dynamics of the four environmental factors [temperature (T), humidity (H), atmospheric pressure (A), and traffic pedestrians (TP)] by PCA using data from previous studies (S1 Table) [15]. These four environmental factors were compressed into PC1, dominated by changes in H/T [Loadings: T(0.62), H(0.647), TP(0.399), A(0.196)], and PC2, primarily reflecting changes in A/TP [Loadings: A(0.825), TP(0.501), H(0.209), T(0.155)] (Fig 2A). The compressed loading values were divided into four distinct groups (G1–G4) based on the date of sample collection (Fig 2B). Notably, each group corresponded to a specific collection date: "2016 May 2" for G1, "2016 June 1" for G2, "2016 July 5" for G3, and "2017 July 15" for G4. Next, we compared the numbers of viable bacteria and particles between groups. While the number of inorganic particles significantly increased from G4 to G1 (Fig 3A), the count of live bacteria was highest in G2, with no other clear pattern (Fig 3B). The results from these graphs were consistent with findings from previous reports involving daily comparisons ([15] (see "Figs 3B and 4A" in this citation). However, in our earlier studies, we did not explicitly confirm that these variations were attributable to a combination of specific environmental factors. Thus, it is now evident that the count of viable bacteria and inorganic particles suspended in the underground walkway varies in response to distinct changes in environmental factors.

### The degree of fit to Glm2 revealed that the dynamics of fine particles can be simply explained by two factors (H/TP), whereas the dynamics of airborne live bacteria are dependent on all four environmental factors (T/H/A/TP)

Next, we employed Glm2 to assess the degree of fit for environmental factors explaining variations in bacterial and particle counts between datasets. Three key indicators ("Residual

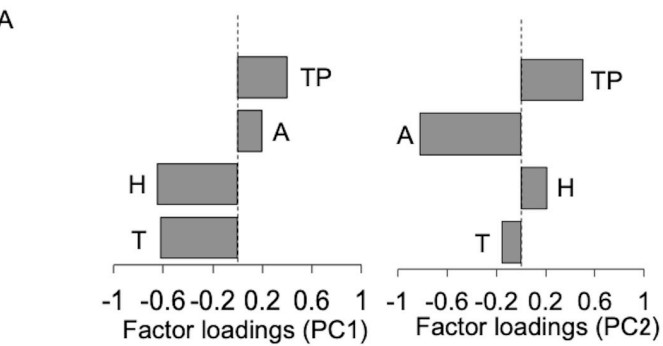

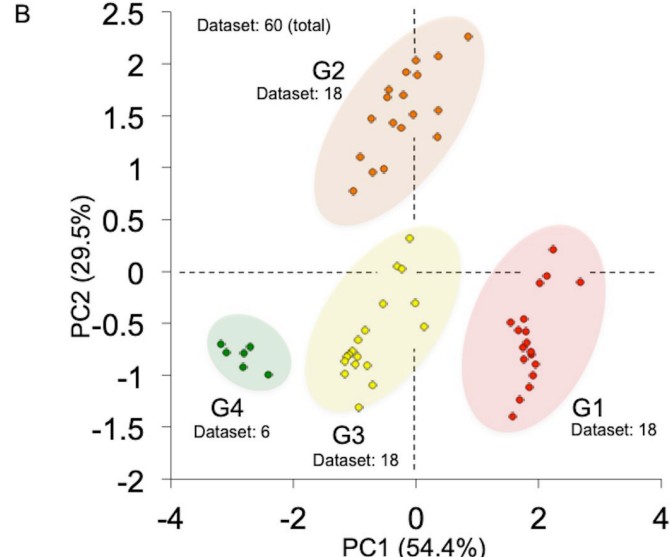

**Fig 2. Characteristic dynamics of the four environmental factors [temperature (T), humidity (H), atmospheric pressure (A), and traffic pedestrians (TP)] as determined by PCA. A.** Bars show the values of factor loadings for PC1 and PC2. Dotted lines show the "Zero" of the values. **B.** PCA plots show the four groups (G1–G4) divided by the distinct weight of environmental factors (Dataset = 60). The compressed loading factors of PC1 (54.4%) and PC2 (29.5%) are [T(0.62), H(0.647), TP(0.399), A(0.196)] and [A(0.825), TP(0.501), H(0.209), T(0.155)], respectively.

Deviance," "AIC," and "logLik") were calculated and compared in two scenarios: "vs null" (no consideration given to environmental factors) and "vs full" (considering all environmental factors). In our analysis, we aimed to identify the smallest combination of environmental factors that closely approximated the fitting value of "full." Remarkably, the three indicators consistently changed in a similar manner, underscoring the accuracy of this model. In terms of the dynamics of particles, as compared with the values of "full," a rapid decrease to the level of "null" was seen in "T" and "H," and the combined value of the two factors ("H"+"A" and "H"+"TP") almost matched the "vs full" value. This result indicated that the dynamics of particles can be easily explained by simple factors alone (Fig 4). By contrast, when examining the dynamics of airborne live bacteria, the situation is more complex. The indicator values gradually decrease to the "null" level, indicating that to understand the dynamics of live bacteria, almost all environmental factors should be considered (Fig 5). We therefore verified its validity through statistical processing using ANOVA with $\chi^2$ distribution. As expected, it was found

A

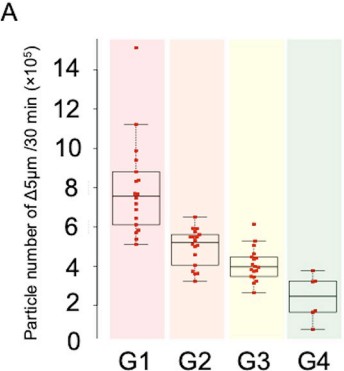

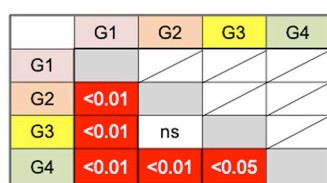

B

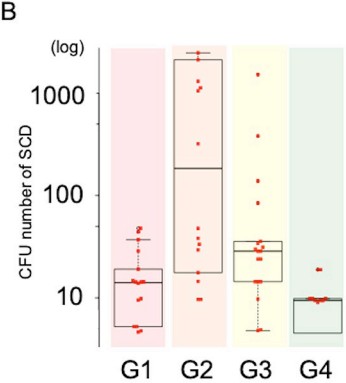

**Fig 3. Airborne live bacteria and particles in the underground space showed different dynamics. A.** Left graph shows the comparison of particles (Δ5: 1–5 μm) suspended in the underground space between groups (G1–G4). Right table shows the results of statistical analysis with the Bonferroni–Dunn test. A *p*-value of < 0.05 (white highlight inside the red frame) shows a statistically significant difference between the two groups. **B.** Left graph shows a comparison of airborne live bacteria in the underground space between the groups (G1–G4). Right table shows the results of statistical analysis. See above (Fig 3A).

that the dynamics of fine particles can be explained in full by only two variables (H+TP) (Fig 6A, see "Red color"), whereas the dynamics of live bacterial counts could not be explained in full by less than the complete set of variables (Fig 6B, see "Red color"). These results clearly indicate that the dynamics of particles and live bacteria are influenced by different environmental factors.

## Changes in bacterial flora are minimal among the four groups (G1–G4) divided by PCA with environmental factors

To explore the potential influence of changes in bacterial flora on the results, PCA was conducted using bacterial flora data from previous studies (S2 Table) [15]. Then, the plots were compared among the four groups distinguished by the impact of environmental factors. The number of OTUs, representing the bacterial load, peaked in G4, suggesting an increase in bacterial abundance from early spring to summer (Fig 7A). However, no discernible alterations in bacterial flora were noted between the groups (Fig 7B and 7C). These results suggested minimal changes in bacterial flora among the four groups (G1–G4), as identified by PCA based on environmental factors.

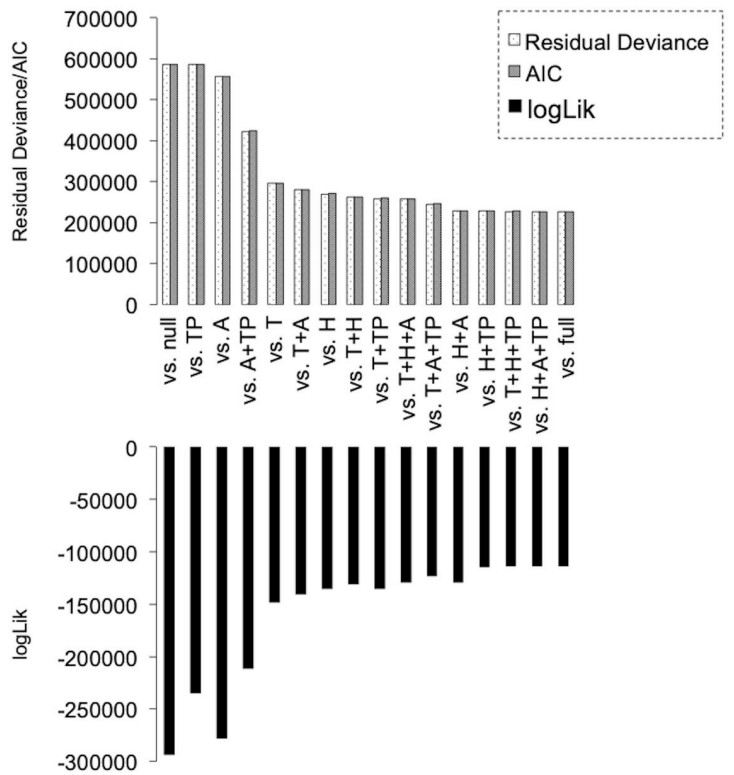

**Fig 4. Degree of fitting to Glm2 revealed that the dynamics of particles can be simply explained by two factors (H/ TP).** Upper graph shows the changes in the values of "Residual Deviance" and "AIC" based on the combination of environmental factors. "vs null" (worst fitting model), negative control. "vs full" (best fitting model), positive control. Higher values indicate poor fit. Lower graph shows the changes in the values of "logLik" based on the combination of environmental factors. "vs null" (worst fitting model), negative control. "vs full" (best fitting model), positive control. Lower values indicate poor fit. "Residual Deviance" and "AIC" indicate the degree of poor fit, and "logLik" indicates the degree of best fit.

## Discussion

We visualized the dynamics of environmental factors using PCA with the previous datasets obtained from the underground walking space [15], and subsequently examined the impact of these dynamics on suspended particles and airborne live bacteria by fitting Glm2 with ANOVA using $\chi^2$. Our previous field survey in this walking space failed to visualize differences in environmental factors affecting floating inorganic particles and the number of viable bacteria [15]. However, our reanalysis newly revealed distinct mathematical patterns in the dynamics of airborne live bacteria compared with inorganic particles. Specifically, while particles exhibit simpler dynamics, the behavior of live bacteria is intricately regulated by multiple environmental factors. Such knowledge is extremely valuable for maintaining a safe and secure public environment with minimal risk of infection from airborne pathogens.

Using PCA, the dynamics of four environmental factors were compressed in two dimensions, and the dataset clearly divided into four groups (G1–G4) based on the sampling date. Temperature and humidity predominantly influenced the X-axis variation, shifting from G1 to G4 as the sampling date moved from May to July, reflecting seasonal changes. Atmospheric pressure and the number of people passing by were the primary contributors to Y-axis changes. Notably, G2, characterized by decreased atmospheric pressure and light rain,

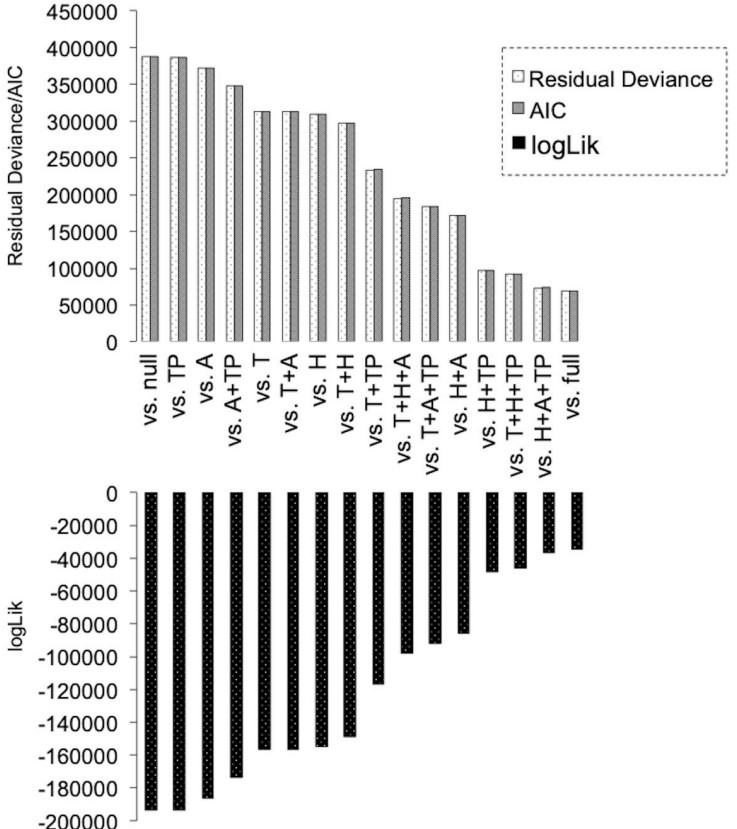

**Fig 5. Degree of fitting to Glm2 revealed that the dynamics of airborne live bacteria can only be explained by multiple factors (T/H/A/TP).** Upper graph shows changes in the values of "Residual Deviance" and "AIC" based on the combination of environmental factors. "vs null" (worst fitting model), negative control. "vs full" (best fitting model), positive control. Higher values indicate poor fit. Lower graph shows changes in the values of "logLik" based on the combination of environmental factors. "vs null" (worst fitting model), negative control. "vs full" (best fitting model), positive control. Lower values indicate poor fit. "Residual Deviance" and "AIC" indicate the degree of poor fit, and "logLik" indicates the degree of best fit. See the legend of Fig 4.

occupied a distinct position from other groups with fair weather [15]. Thus, the PCA plot effectively captured seasonal variations with the flow of pedestrians, clearly delineating the dataset into the four groups reflecting the distinct weighting of environmental factors.

To verify the PCA results, we used a fitting model (Glm2) with ANOVA using $\chi^2$. This model, which follows a Poisson distribution, allows for an optimal probability distribution when the data are discrete values with no linear variations and the upper limit is unpredictable [19]. In other words, the model allows the probability distribution of environmental factors to be visualized for each combination to explain the changes in the number of bacteria and particles, and their differences can be easily determined by comparing three indicators ("Deviance," "logLik," and "AIC"). Moreover, the degree of fitting for each combination of predictors (environmental factors) can be simply calculated for significance using an approximate calculation method (ANOVA using $\chi^2$).

As expected, PCA revealed that fine particles varied significantly in value between the groups (G1–G4). Specifically, the number of fine particles ($\Delta 5$: 1–5 µm) with values $>10^5$ gradually decreased from G4 to G1. This change may be related to the fact that the temperature

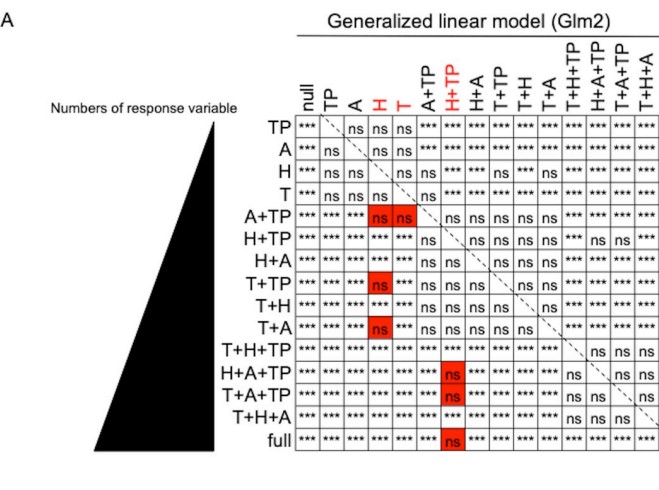

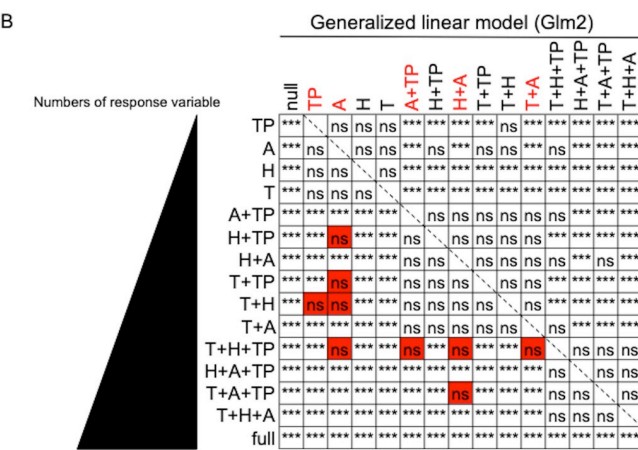

**Fig 6. Statistical analysis using ANOVA with $\chi^2$ distribution verified the comparison results for the degree of fitting to Glm2.** A significance test was performed to determine the difference in the degree of fitting to Glm2 for each combination of explanatory variables (T/H/A/TP) to explain the response variable (**A:** dynamics of particles, **B:** dynamics of airborne live bacteria). ***, statistically significant with $p < 0.05$. "ns", not significant. Red color shows that a combination of fewer explanatory variables is comparable to a combination of more explanatory variables.

and humidity of this underground walking space increased from spring to summer, and this finding was consistent with those of other studies [20–22]. The data for G4 comprised the least number of pedestrians late at night and early in the morning, indicating that the quantity of suspended particles in the space is also affected by the number of pedestrians. In fact, studies have reported that the number of people present in elevators or subway stations influences the dynamics of particles suspended in indoor environments [23, 24].

Interestingly, PCA and a comparative analysis among the four visualized groups suggested that the number of particles suspended in underground walking spaces varies depending on temperature, humidity, and the number of pedestrians. To verify this, we compared the degree of fitting using the Glm2 model. As expected, it was clear that although the dynamics of particles can be simply explained by temperature and humidity alone, they are more comprehensively explained by a combination of humidity and the number of pedestrians. Thus, these results suggest that the fluctuations in the number of particles suspended in this underground

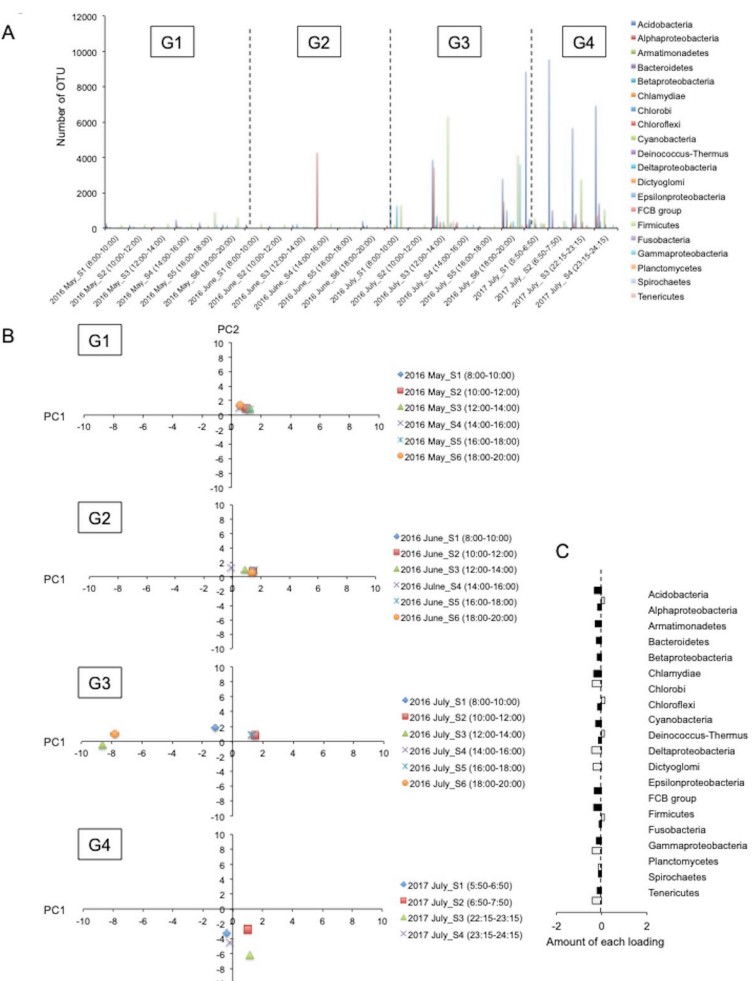

**Fig 7. Changes in bacterial flora were minimal among the four groups (G1–G4) divided by PCA with environmental factors. A.** Graph showing the number of OTUs (Dataset = 22), representing the bacterial load, among the groups (G1–G4) divided by PCA with environmental factors. **B.** Plots showing changes in the dynamics of the bacterial flora using PCA for each group (G1–G4). PC1, 36.1%. PC2, 57.8%. **C.** Graph showing the loading amounts (PC1 and PC2) of each phylum of bacteria.

walking space can be explained by two environmental factors, namely humidity and the number of pedestrians.

By contrast, the dynamics of the number of airborne live bacteria in the underground walking space showed a completely different pattern from that of fine particles. Specifically, PCA with comparative analysis among the four groups (G1–G4) clearly showed that the number of live bacteria significantly increased in G2. This finding suggests that the dynamics of live bacteria were dependent on a certain temperature and humidity with an increasing number of pedestrians and decreasing atmospheric pressure. Similarly, the model analysis of Glm2 with $\chi^2$ revealed that the dynamics could not be explained by specific environmental factors or simple combinations. Thus, it is apparent that all of the environmental factors considered in this study are required to explain the dynamics of live airborne bacteria in the underground space.

It is unclear why environmental factors affecting the dynamics of particles in underground walkways differ from those influencing the count of live bacteria. Notably, there was a

significant difference in the total number of airborne particles versus bacteria, with particles showing 10,000 times higher values. Because most airborne bacteria adhere to some form of inorganic fine particles [25, 26], this implies that the proportion of fine particles with bacteria attached is small. In other words, although studies are being conducted on the relationship between the number of fine particles and the number of airborne bacteria in public environments [27–29], particle count alone may not be an appropriate indicator of the number of airborne bacteria in public spaces.

The explanation as to why the number of airborne live bacteria in the underground space significantly increased under such limited environmental conditions remains unclear; however, the following observations may provide some insight. First, the survival of bacteria, such as *Escherichia coli* or *Staphylococcus aureus*, on dry surfaces is enhanced in low temperature and low humidity environments [30]. Second, the survivability of bacteria attached to a dry surface heated to the level of human skin (~37˚C) was significantly lower than when the surface was not heated [31, 32]. Third, given that no change in bacterial flora was observed between the groups (G1–G4), the increase in the total number of bacteria itself has an impact. Thus, moderate temperature and humidity may have a significant impact on the survival of airborne bacteria attached to particles.

This study, conducted in the underground walking space in Sapporo, has some limitations. First, to ensure the universality of these results, similar studies in other public environments are crucial. Second, the count of live bacteria was surprisingly small compared with bacterial flora analysis results. This suggests that additional investigation is needed into the validity of the culture method and the potential presence of "viable but non-culturable" bacteria. Third, although environmental factors were visualized using a fitting model, it is essential to verify whether the new measurements align with the expected probability distribution.

In summary, we newly found that the dynamics of airborne live bacteria in the underground walkway differ from those of particles and are intricately regulated by multiple environmental factors. This study stands out as one of the few to mathematically and statistically visualize the environmental factors influencing the number of airborne bacteria in public spaces, contributing to the enhancement of public health in urban settings.

## Supporting information

**S1 Table. Dataset reused from our previous study (Dataset = 60) [15].** The datasets include "temperature (T)," "humidity (H)," "atmospheric pressure (A)," "traffic pedestrians (TP)," "number of inorganic particles (Δ5: 1–5 μm)," and "number of live airborne bacteria". (PDF)

**S2 Table. Airborne bacterial flora dataset obtained from our previous study (Dataset = 22) [15].** Values show the number of OTUs. (PDF)

## Acknowledgments

We thank Edanz, a professional proofreading company, (https://jp.edanz.com/ac), for editing a draft of this manuscript.

## Author Contributions

**Conceptualization:** Hiroyuki Yamaguchi.

**Data curation:** Hiroyuki Yamaguchi, Torahiko Okubo, Eriko Nozaki, Takako Osaki.

**Formal analysis:** Eriko Nozaki, Takako Osaki.

**Funding acquisition:** Hiroyuki Yamaguchi.

**Investigation:** Hiroyuki Yamaguchi, Torahiko Okubo, Eriko Nozaki, Takako Osaki.

**Supervision:** Hiroyuki Yamaguchi.

**Writing – original draft:** Hiroyuki Yamaguchi.

**Writing – review & editing:** Hiroyuki Yamaguchi, Eriko Nozaki, Takako Osaki.

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
