## [Decision Letter · Decision Letter 0]

9 Jan 2024

PONE-D-23-40119Differential impact of environmental factors on airborne live bacteria and inorganic particles in an underground walkwayPLOS ONE

Dear Dr. Yamaguchi,

Thank you for submitting your manuscript to PLOS ONE. After careful consideration, we feel that it has merit but does not fully meet PLOS ONE’s publication criteria as it currently stands. Therefore, we invite you to submit a revised version of the manuscript that addresses the points raised during the review process.

We look forward to receiving your revised manuscript.

Kind regards,

Sara Hemati

Academic Editor

PLOS ONE

Journal Requirements:

5. Thank you for stating the following financial disclosure: " This study was funded by a grant-in-aid for scientific research, KAKENHI (grant number

20K20613 to HY)."

6. We note that your Data Availability Statement is currently as follows: "All relevant data are within the manuscript and its Supporting Information files."

7. Thank you for stating the following in the Acknowledgments Section of your manuscript: “We are grateful to the South African Medical Research Council and National Research Foundation for financial support.”

Additional Editor Comments:

Manuscript Number: PONE-D-23-40119

Differential impact of environmental factors on airborne live bacteria and inorganic particles in an underground walkway

Dear Dr. Yamaguchi,

I hope this email finds you well.

Thank you for submitting your manuscript to PLOS ONE. After careful consideration, we feel that it has merit but does not fully meet PLOS ONE’s publication criteria as it currently stands. Therefore, we invite you to submit a revised version of the manuscript that addresses the points raised during the review process.

We look forward to receiving your revised manuscript.

Kind regards,

Sara Hemati

Academic Editor

PLOS ONE

Reviewers' comments:

Reviewer's Responses to Questions

**Comments to the Author**

1. Is the manuscript technically sound, and do the data support the conclusions?

Reviewer #1: Yes

Reviewer #2: Partly

2. Has the statistical analysis been performed appropriately and rigorously? 

Reviewer #1: I Don't Know

Reviewer #2: Yes

3. Have the authors made all data underlying the findings in their manuscript fully available?

Reviewer #1: Yes

Reviewer #2: Yes

4. Is the manuscript presented in an intelligible fashion and written in standard English?

Reviewer #1: Yes

Reviewer #2: No

5. Review Comments to the Author

Reviewer #1: The difference between this article and your article September 18, 2017 is not significant in terms of results. Was there a special reason for presenting this article? It is better to explain the necessity of doing research.

Reviewer #2: Dear Authors,

The manuscript was fully reviewed. The comments in the manuscript. The manuscript needs a major revision before any consideration. The view of the study is good but some concerns exist. Best

6. PLOS authors have the option to publish the peer review history of their article (what does this mean?). If published, this will include your full peer review and any attached files.

Reviewer #1: No

Reviewer #2: No

---

## [Author Response · Author response to Decision Letter 0]

28 Feb 2024

REPLIES TO REVIEWER’S COMMENTS

Ref. ID: PONE-D-23-40119

Differential impact of environmental factors on airborne live bacteria and inorganic particles in an underground walkway

Responses to Editor Comments:

Comment 1: Thank you for submitting your manuscript to PLOS ONE. After careful consideration, we feel that it has merit but does not fully meet PLOS ONE’s publication criteria as it currently stands. Therefore, we invite you to submit a revised version of the manuscript that addresses the points raised during the review process.

Response: We have incorporated all reviewer comments and journal requirements into the revised version according to the editor's instructions.

Comment 2: Please submit your revised manuscript by Feb 09 2024 11:59PM. If you will need more time than this to complete your revisions, please reply to this message or contact the journal office at plosone@plos.org. Response: According to the instructions, we submitted the revised manuscript within the extended deadline (the end of March).

Comment 3: A rebuttal letter that responds to each point raised by the academic editor and reviewer(s). You should upload this letter as a separate file labeled 'Response to Reviewers'.

Response: According to the instructions, we uploaded the response letter s a separate file labeled 'Response to Reviewers'..

Comment 4: A marked-up copy of your manuscript that highlights changes made to the original version. You should upload this as a separate file labeled 'Revised Manuscript with Track Changes'.

Response: According to the comment, we highlighted the corrections in red color and uploaded the revised manuscript as a separate file labeled 'Revised Manuscript with Track Changes'.

Comment 5: An unmarked version of your revised paper without tracked changes. You should upload this as a separate file labeled 'Manuscript'.

Response: Yes, according to the instructions, we uploaded a unmarked revised manuscript as a separate file labeled 'Manuscript'.

Comment 6: Response: Since this research is supported by a grant-in-aid for scientific research, KAKENHI (grant number 20K20613 to HY), the following sentence has been added in the section 'financial disclosure'. 'This research is supported by a grant-in-aid for scientific research, KAKENHI (grant number 20K20613 to HY). The funders had no role in study design, data collection and analysis, decision to publish, or preparation of the manuscript'. According to the comment, we wrote about the corrections in the cover letter.

Comment 7: Guidelines for resubmitting your figure files are available below the reviewer comments at the end of this letter.

Response: We have verified that the figures and tables follow the guidelines. No revisions to figures or tables were requested from the reviewers.

Comment 7: If applicable, we recommend that you deposit your laboratory protocols in protocols.io to enhance the reproducibility of your results. Protocols.io assigns your protocol its own identifier (DOI) so that it can be cited independently in the future. 

Response: Thank you for recommending the deposit of protocols used this study to 'protocols.io'. On the other hand, the R commands used this study is not applicable as it is a general one. 

Responses to Journal Requirements:

Comment 1: Please ensure that your manuscript meets PLOS ONE's style requirements, including those for file naming. 

Response: We have confirmed that it matches the journal style of PLOS ONE, including the file name.

Comment 2: Did you know that depositing data in a repository is associated with up to a 25% citation advantage? If you’ve not already done so, consider depositing your raw data in a repository to ensure your work is read, appreciated and cited by the largest possible audience. You’ll also earn an Accessible Data icon on your published paper if you deposit your data in any participating repository.

Response: Thank you for introducing us to 'Repository'. On the other hand, there is no huge amount of metadata this time, so we don't think it applies. On the other hand, all the raw data used have been summarized in a table as supporting information so that readers can check the reproducibility.

Comment 3: Please note that PLOS ONE has specific guidelines on code sharing for submissions in which author-generated code underpins the findings in the manuscript. In these cases, all author-generated code must be made available without restrictions upon publication of the work. Please review our guidelines at https://journals.plos.org/plosone/s/materials-and-software-sharing#loc-sharing-code and ensure that your code is shared in a way that follows best practice and facilitates reproducibility and reuse.

Response: Thank you for your advice. The R commands we used in this paper are general and do not include any proprietary code. Also, all codes are disclosed in the text, so anyone can perform similar analysis.

Comment 4: We note that the grant information you provided in the ‘Funding Information’ and ‘Financial Disclosure’ sections do not match. When you resubmit, please ensure that you provide the correct grant numbers for the awards you received for your study in the ‘Funding Information’ section.5. Thank you for stating the following financial disclosure: " This study was funded by a grant-in-aid for scientific research, KAKENHI (grant number 20K20613 to HY)." Please state what role the funders took in the study. If the funders had no role, please state: "The funders had no role in study design, data collection and analysis, decision to publish, or preparation of the manuscript." If this statement is not correct you must amend it as needed. Please include this amended Role of Funder statement in your cover letter; we will change the online submission form on your behalf.

Response: Thank you for your advice. According to the comment, we have checked the funding information, and have declared the following sentence 'The funders had no role in study design, data collection and analysis, decision to publish, or preparation of the manuscript' into the Financial Disclosure' of the revised manuscript.

Comment 6: We note that your Data Availability Statement is currently as follows: "All relevant data are within the manuscript and its Supporting Information files." Please confirm at this time whether or not your submission contains all raw data required to replicate the results of your study. Authors must share the “minimal data set” for their submission. PLOS defines the minimal data set to consist of the data required to replicate all study findings reported in the article, as well as related metadata and methods. For example, authors should submit the following data:

Authors do not need to submit their entire data set if only a portion of the data was used in the reported study. If your submission does not contain these data, please either upload them as Supporting Information files or deposit them to a stable, public repository and provide us with the relevant URLs, DOIs, or accession numbers. For a list of recommended repositories, please see https://journals.plos.org/plosone/s/recommended-repositories. If there are ethical or legal restrictions on sharing a de-identified data set, please explain them in detail (e.g., data contain potentially sensitive information, data are owned by a third-party organization, etc.) and who has imposed them (e.g., an ethics committee). Please also provide contact information for a data access committee, ethics committee, or other institutional body to which data requests may be sent. If data are owned by a third party, please indicate how others may request data access.

Response: We have confirmed that all previously published paper data used for R analysis is listed in the supporting information.

Comment 7: Thank you for stating the following in the Acknowledgments Section of your manuscript: “We are grateful to the South African Medical Research Council and National Research Foundation for financial support.” We note that you have provided funding information that is currently declared in your Funding Statement. However, funding information should not appear in the Acknowledgments section or other areas of your manuscript. We will only publish funding information present in the Funding Statement section of the online submission form.

Response: According to the comment, we corrected it.

To Reviewer 1: 

Comment 1 (line 38): in length? in width?

Response: Sorry for not explaining enough. It is a straight walking space with a length of 520m. we corrected it

Comment 2 and 3 (line 62~): Adding: inorganic particles and environmental factors

Response: According to the comment, We added these to the keywords of the revised manuscript.

Comment 4 (line 64~): The difference between this article and your article September 18, 2017 is not significant in terms of results. Was there a special reason for presenting this article? It is better to explain the necessity of doing research.

Response: We apologize for the lack of clarity in addressing your points. The novelty of this paper lies in its use of a mathematical model to clarify the interconnectedness of environmental factors. This elucidation was previously unclear based on field data alone. We've added an explanation to this point in the introduction of the revised manuscript, as suggested in the comment.

To Reviewer 2: 

Comment 1: The whole text needs proofreading.

Response: According to the comment, the revised paper has been edited in English by Edanz, an English editing company, as suggested in the comment.

Comment 2: The novelty should be highlighted.

Response: The novelty of this paper lies in its use of a mathematical model to clarify the interconnectedness of environmental factors. This elucidation was previously unclear based on field data alone. We've added an explanation to this point in the abstract and introduction of the revised manuscript, as suggested in the comment.

Comment 3: Here you should mention the place and time of conducting the study.

Response: According to the comment, we mentioned the place and time of conducting the study in the abstract of the revised manuscript.

Comment 4: Introduction needs improvement. These papers are useful:

1.https://doi.org/10.1016/j.chemosphere.2023.140627

2.https://doi.org/10.1038/s41598-023-37647-3

3.https://doi.org/10.1016/j.envpol.2023.121854

Response: We have incorporated the suggested papers into the introduction of the revised manuscript, updating the reference numbers accordingly throughout.

Comment 5 (Materials and Methods): You should re-design this section as it can be repeated easily as a recipe. 

Following the suggestion, we have redesigned the section, consolidating the command lines to make them easier for readers to repeat.

Comment 6 (Discussion): You should discuss your findings with new findings of relevant studies.

Response: Following comments, we have added an explanation of the novelty and usefulness of this study to the discussion of the revised manuscript.

---

## [Editor Report · Decision Letter 1]

6 Mar 2024

Differential impact of environmental factors on airborne live bacteria and inorganic particles in an underground walkway

PONE-D-23-40119R1

Dear Dr. Yamaguchi,

We’re pleased to inform you that your manuscript has been judged scientifically suitable for publication and will be formally accepted for publication once it meets all outstanding technical requirements.

Kind regards,

Sara Hemati

Academic Editor

PLOS ONE
---

## [Editor Report · Acceptance letter]

12 Mar 2024

PONE-D-23-40119R1 

PLOS ONE

Dear Dr. Yamaguchi, 

I'm pleased to inform you that your manuscript has been deemed suitable for publication in PLOS ONE. Congratulations! Your manuscript is now being handed over to our production team.

Kind regards, 

on behalf of

Dr. Sara Hemati 

Academic Editor

PLOS ONE